# Trunk kinematics and motor unit behavior during different loads and speeds in individuals with and without aberrant movement patterns during active forward bending: A cross-sectional study

**Peemongkon Wattananon** [ID] [1]*, **Sasithorn Kongoun**[1], **Katayan Klahan**[1], **Sheri P. Silfies** [ID] [2], **John R. Gilliam**[2], **Jim Richards** [ID] [3]

**1** Spine Biomechanics Lab, Faculty of Physical Therapy, Mahidol University, Nakhon Pathom, Thailand, **2** Applied Neuromechanics Lab, Arnold School of Public Health, University of South Carolina, Columbia, SC, United States, **3** Allied Health Research Unit, University of Central Lancashire, Preston, United Kingdom

* peemongkon.wat@mahidol.ac.th

## Abstract

### Background

Instability catch (IC) during active forward bending is an aberrant movement pattern observed in patients with low back pain. Increasing load and speed may show different responses in kinematics and motor unit behavior including peak amplitudes (pAMP) and mean firing rates (mFR).

### Objectives

This study aimed to compare kinematic patterns under different loads and speeds and explored the motor unit behavior in individuals with and without IC.

### Methods

17 participants were classified as having IC and 10 participants were classified as having no IC from clinical observations. Inertial measurement units were used to quantify kinematic parameters, and decomposition electromyography (dEMG) was used to investigate motor unit behavior. Participants performed 2 sets of 1-minute forward bending under low load and low speed (LL), high load and low speed (HL), and low load and high speed (LH) conditions.

### Results

Significant between-group differences ($P < 0.05$) were found in kinematic parameters. Significant within-group changes ($P < 0.05$) were found between the LL and HL conditions for all kinematic parameters in individuals with IC. Individuals without IC demonstrated

**Data availability statement:** All relevant data are within the manuscript and its Supporting Information files.

**Funding:** This study was funded by the National Research Council of Thailand (N42A650360) (awarded to PW), the Mahidol University (MU Talent Program) (awarded to PW), and the Faculty of Physical Therapy, Mahidol University (Research Assistant Fund) (awarded to KK). The funders did not play any role in the study design, data collection and analysis, decision to publish, or preparation of the manuscript.

**Competing interests:** The authors have declared that no competing interests exist.

significant within-group changes ($P<0.05$) between LL and LH in mFR, while individuals without IC showed changes in both pAMP and mFR.

## Conclusion

These kinematic parameters may represent IC. Changes in motor unit behavior suggest that individuals with and without IC used different strategies to perform this task. Clinicians may consider varying the speed of movement to challenge the trunk neuromuscular control system and design interventions to address motor unit firing rate.

## 1. Introduction

Clinical observation of movement control and coordination during active forward bending of the trunk is one critical part of the physical examination for patients with low back pain (LBP) [1–3]. Aberrant movement patterns during active forward bending cause shear forces and suboptimal tissue loading at the spine resulting in an increased risk of tissue damage [1,4,5]. Aberrant movement patterns have consistently been identified in individuals with a history of LBP, and investigators have speculated that this could be due to unresolved lumbar multifidus (LM) muscle dysfunction [1,4,6]. Evidence demonstrates the LM does not show spontaneous recovery after an episode of LBP [7–9], with previous work demonstrating a persistent impairment of LM activation using ultrasound imaging in individuals who were in remission following an episode of LBP [8]. Impaired LM activation could compromise lumbar stability and result in an instability catch (IC) during active forward bending. Instability catch (IC) is defined as a momentary quiver, vibration, or shake seen in the lumbar region [1].

Kinematic studies using a dynamic systems approach have characterized IC as a sudden deceleration and acceleration represented by the frequency of local minimum occurrences [10,11]. However, observed IC in the clinical setting could be triggered by the amplitude of shaking and/or the duration of sudden deceleration and acceleration. Therefore, including kinematic parameters of amplitude and time during deceleration and acceleration may further characterize IC which should result in better representation of this construct. Peak-to-peak (amplitude) and area (amplitude versus time) measures are viable candidates for kinematic parameters to represent IC identified by clinical observation. In addition, changes in the load and speed of the active forward bend may cause changes in neuromuscular demands and result in further changes in the movement patterns [12,13]. Changes in movement patterns, especially IC, might be caused by motor unit recruitment patterns. However, evidence to support motor unit behavior underlying IC is still limited.

Typically, healthy individuals initially recruit smaller motor units [14]. As neuromuscular demands increase such as increases in speed, so do motor unit firing rates [15]. If demands increase past the capacity of smaller motor units, additional larger motor units are recruited to match the task requirements [14,15]. Decomposition electromyography (dEMG) is a new technology that can be used to investigate motor unit behavior by decomposing the EMG signal into individual motor unit action potential trains from which peak amplitude (pAMP) and mean firing rates (mFR) may be extracted [16–18].

Therefore, this study aimed to compare kinematic patterns during active forward bending including number of acceleration zero-crossings (NUM), peak-to-peak acceleration amplitude (P2P) and area under acceleration graphs (AUC) under different load and speed conditions. Further, we aimed to explore the motor unit behavior (pAMP and mFR) underlying aberrant

movement. We hypothesized that increased load and speed would require greater motor unit recruitment resulting in less aberrant movement.

## 2. Methods

### 2.1 Study design

This was a cross-sectional study using a two-factor mixed model design to determine the effects of speed and load on trunk kinematics and LM motor unit behavior between people with and without aberrant movement patterns. This study was approved by the Mahidol University Institutional Review Board (COA No. 2022/118.0711). This study followed the principles of the Declaration of Helsinki. Informed consent for publication of identifying information/images in an online open-access publication has also been obtained. Data were collected from December 2022 to November 2023.

### 2.2 Participants

A convenience sample of asymptomatic participants were recruited from the University and surrounding areas. Inclusion criteria were age between 20–40 years, and currently symptom free. Prior episode of LBP was not an exclusion criterion. Participants were excluded if they had definitive neurologic signs including weakness or numbness in the lower extremity, previous spinal surgery, diagnosed osteoporosis, spinal stenosis, inflammatory joint disease, or systemic disease, and a BMI greater than $30\,kg/m^2$. All participants provided written informed consent before data collection. This sample was part of an ongoing study aimed to explore the potential effects of different interventions on motor unit behavior. A previous study found differences in the number of sudden deceleration and acceleration (zero-crossings) representing IC (judder) between individuals with IC ($11.7 \pm 4.6$ occurrences) and without IC ($6.7 \pm 2.5$ occurrences) [11]. These data were used to determine the sample size required using an 80% power and an alpha of 5%. A total of 20 participants, at least 10 participants per group, were determined necessary to detect differences between groups.

### 2.3 Instruments and measures

Two Inertial Measurement Unit (IMU) sensors (Trigno Avanti, Delsys Inc., MA, USA) were attached to the lumbar (L1) and sacral (S2) spinous processes. Angular velocity data were recorded directly from the gyroscopes of the IMUs during an active forward bend movement at 370 Hz. As these were carefully placed on anatomical segments and no further parameters were calculated, these required no additional calibration process [19,20]. The sensor axes were used to define the planes of movement in rotation about X (flexion/extension), Y (rotation), and Z (lateral bending) of the lumbar and sacral segments in which flexion, rotation to the right, and lateral bend to the left were considered positive directions (Fig 1A). This method has been previously used to explore lumbopelvic movements [19,20], and angular velocity measures have shown excellent test-retest reliability to assess movement pattern consistency (coefficient of multiple determination = 0.85) [20] and sensitivity changes between conditions [21].

Two dEMG sensors (Trigno Galileo, Delsys Inc., MA, USA) were attached bilaterally to the lumbar multifidus (2 cm lateral to L4 spinous process) with the reference attached over the iliac crests. Each sensor collected four channels of differential EMG data at 2222 Hz from 4 protruding blunted pins (0.5 mm in diameter) with 5-mm inter-pin space. EMG data were amplified with a gain of 1000 and filtered at a band-pass of 20–450 Hz [15,18,22]. This system has been previously utilized to explore the motor unit behavior of back muscles in healthy individuals [22,23], and the effect of increases in neuromuscular demand during dynamic movements [15].

## 2.4 Procedure

Demographic data including age, sex, weight, height, and BMI were recorded. Participants were asked to perform 3 repetitions of active forward bending at their most comfortable pace, while a researcher observed their movement and rated the presence or absence of an aberrant movement pattern. Movement pattern observation was performed by a physical therapist with 10 years of clinical experience in assessment and treatment of LBP who did not participate in the data analysis. Participants were rated as having IC if they had obvious shaking in their lumbar area during forward bending. These ratings were used to classify participants to one of two groups with either the presence or absence of IC with pilot data demonstrating a moderate inter-rater reliability of clinical classification (kappa = 0.52).

After classification participants were asked to expose their lumbopelvic area (L1 to S2). Skin preparation was performed using a 70% alcohol swab prior to IMU and dEMG sensor attachment (Fig 1A). Data collection was performed using EMGworks 4.7.3 (Delsys Inc., MA, USA) by two researchers who were blinded to the group assignment. Participants were asked to relax in the prone position and baseline noise was assessed to ensure the value was less than ±10 microvolts for the dEMG data. If the baseline noise was greater than ±10 microvolts this process was repeated until this baseline noise level was achieved.

Two speeds and two loads were considered. The two speeds were controlled by asking the participants to keep in time with a metronome set at 30 and 50 beats per minute for the downward and upward movements, thus giving a complete movement rate of 15 and 25 repetitions a minute, which has been previously used to assess motor unit behavior and kinematics around the knee [15], and also approximated to the mean velocity of participants performing the movement at a self-selected comfortable pace and the maximum pace that participants could consistently keep in time. Two loads 5% and 10% of body weight were used during the two speeds using kettle bells held in front of the body with arms straight. The 5% of body weight load represented tasks of daily living, and the 10% of body weight load represented the maximum weight that participants could perform for 1 min of repeated forward bending without fatigue. The combination of speed and load were used to create the conditions; low

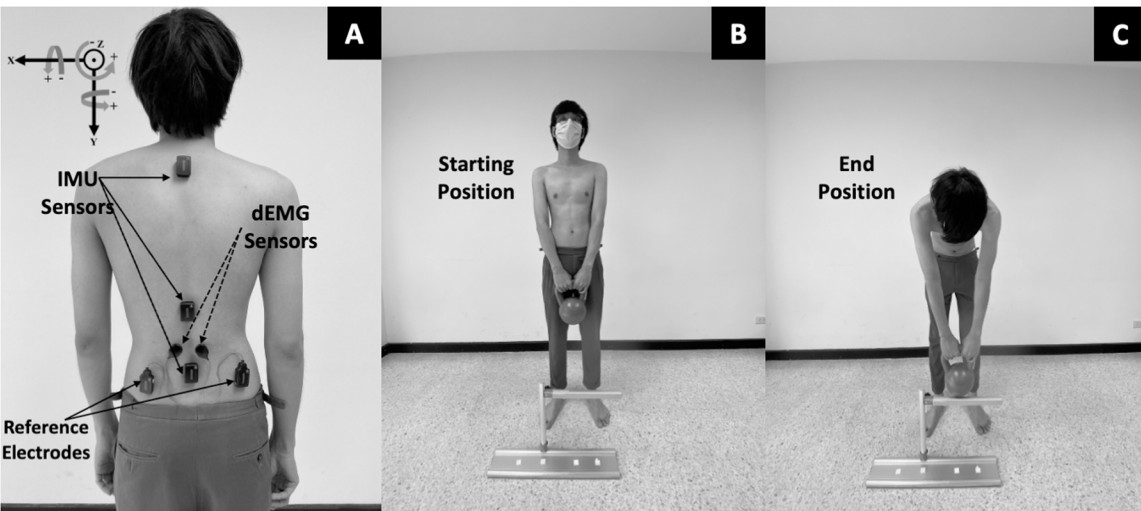

**Fig 1. Inertial measurement unit (IMU) and decomposition electromyography (dEMG) sensor locations (A) and task starting (B) and end (C) positions.**

speed and low load (LL), low speed and high load (LH), and high speed and low load (HL). We did not include a high speed and high load condition because participants were unable to consistently perform forward bend throughout 1-minute task in our pilot study. Therefore, we included only LL, LH and HL conditions for this study. The participants were asked to perform 2 sets of forward bends to 45-degrees of lumbar flexion, which was standardized by adjusting the height of a target bar (Fig 1B and 1C), each for 1 minute with a 5-minute rest between sets. The three conditions (LL, LH, and HL) were performed in a random sequence. We did not include high speed and high load in our study because our pilot work found that most participants reported muscle fatigue after completing this condition. IMU and dEMG data were simultaneously collected during each condition. All relevant data are included in the manuscript and its supporting information files.

## 2.5  Data reduction

Kinematic data reduction was performed using a custom LabVIEW program (National Instruments, TX, USA). All IMU data were filtered using a second order lowpass Butterworth filter at 20 Hz. Assessments of sagittal plane aberrant movements during active forward bending were evaluated from the flexion/extension angular velocity data from the lumbar sensor. Start and stop events (neutral position to targeted position) were identified using 5% of maximum lumbar angular velocity as a cut-off point. Mean angular velocity (MV), peak angular velocity (PV), as well as number of acceleration zero-crossings (NUM), peak-to-peak acceleration amplitude (P2P), and total area under sudden deceleration and acceleration curves (AUC) were derived. These parameters were used for further statistical analysis.

For dEMG data processing, no filtering was required. NeuroMap software version XXXX (Delsys, Inc., Boston, USA) was used to decompose the EMG signals into individual motor units using an artificial intelligence algorithm [18]. Neuromap Explorer (Delsys, Inc., Boston, USA) was then used to extract the peak motor unit amplitude (pAMP) and mean firing rates (mFR) that had an accuracy of 80% or greater, which is supported by De Luca et al. [16], who demonstrated that 80% is appropriate for identifying a comparable number of motor unit action potential trains during dynamic, cyclic tasks.

## 2.6  Statistical analysis

Statistical analysis was performed using SPSS version 21 (IBM Corp., NY, USA). An independent t-test was used to determine the difference in age, BMI, mean angular velocity (for each condition), while a chi-square test was performed to determine the difference in sex proportion between groups.

The distribution of data was tested using Shapiro-Wilk tests, and all kinematic and dEMG data were normally distributed. Kinematic parameters included mean angular velocity (MV), peak angular velocity (PV), number of acceleration zero-crossings (NUM), peak-to-peak acceleration amplitude (P2P), and total area under sudden deceleration and acceleration curves (AUC). dEMG parameters include peak motor unit amplitude (pAMP) and mean firing rates (mFR). Before main data analysis, we used data from 2 sets of 1-minute forward bends to calculate test-retest reliability and standard error of measurement (SEM) for each parameter to strengthen our internal validity. We further used the SEM to determine whether within and between-group differences resulted from measurement error or not. We used a two-factor mixed model ANOVA with post-hoc least significant difference (LSD) and SEM to determine interaction and main effects of group (presence and absence of IC) and condition (LL, LH and HL). Effect sizes of two-factor mixed model ANOVA were reported using partial eta squared (partial $\eta^2$) and interpreted as small (0.01), medium (0.06), and large (0.14), while

effect sizes of pairwise comparison were reported using Cohen's d and interpreted as small (0.3), medium (0.5), and large (0.8) [24].

In addition, dEMG (pAMP and mFR) data from individual participants were used to construct profiles indicating increased or decreased response to changes in speed and load in each parameter using LL as the reference value. The frequency and percentage of participants who demonstrated an increase from their reference dEMG parameters or decrease from their kinematic parameters in response to increased speed or load was calculated for each parameter. These percentages were then used to characterize potential neuromuscular strategies for individuals with and without IC.

## 3. Results

Twenty-seven individuals were recruited who were classified as either having IC or not during active forward bending. Ten individuals were classified as having no IC (mean age 24.0 years, 4 females, and BMI 23.3 kg/m$^2$) and 17 individuals were classified as having IC (mean age 22.4 years, 10 females, and BMI 21.4 kg/m$^2$). Demographic and MV data are presented in Table 1. No significant differences were found between individuals with and without IC for age, sex, BMI, or MV.

For dEMG parameters, data from one participant could not be decomposed due to a technical problem, this individual was excluded from the statistical analysis. Our dataset also demonstrated no significant differences in dEMG parameters between left and right sides; therefore, we used side-to-side averaged values for statistical analysis. Although sex could potentially affect dEMG parameters, we did not find any interactions between sex and other parameters in our dataset which was consistent with another dEMG study [15].

Test-test reliability for kinematic parameters (NUM, P2P, and AUC) showed moderate to excellent (ICC$_{2,k}$ = 0.95, 0.72, and 0.91), respectively. SEM values were 0.7 occurrences, 0.35 deg/sec, and 6.03 units, respectively. Test-test reliability for dEMG parameters (pAMP and mFR) were uniformly excellent (ICC$_{2,k}$ = 0.96 and 0.93, respectively). SEM values were 8.12 microvolts, and 0.37 pps, respectively.

Table 2 demonstrates the means, standard errors, and effect sizes for interaction (Group×Condition) and main effect of group and condition for kinematics and motor unit behavior. No significant interactions ($P > 0.05$) were seen between group and condition, but significant main effects of condition ($P < 0.05$) with large effect sizes were seen for all parameters, except pAMP. In addition, significant main effects of group ($P < 0.05$) with large effect sizes were seen for P2P and AUC, while there was a trend ($P = 0.053$) with large effect size for NUM.

**Table 1. Demographic data and performance for each condition.**

| Parameter | Negative (n = 10) Mean (SD) | Positive (n = 17) Mean (SD) |
|---|---|---|
| Age (year) | 24.0 (4.6) | 22.4 (1.3) |
| Sex (male/female) | 6/4 | 7/10 |
| BMI (kg/m$^2$) | 23.3 (3.4) | 21.4 (1.8) |
| LL_MV (deg/sec) | 49.50 (9.32) | 57.44 (11.27) |
| LH_MV (deg/sec) | 48.98 (10.39) | 55.05 (14.13) |
| HL_MV (deg/sec) | 65.36 (12.08) | 72.15 (12.38) |

BMI = body mass index; LL = low speed and low load; LH = low speed and high load; HL = high speed and low load; MV = mean angular velocity; SD = standard deviation

**Table 2. Mean and standard error for interaction (ABM × Condition) and main effect of group and condition for kinematics and motor unit behavior.**

| Parameter | Interaction (ABM×Condition) | | Main effect | | | | | | | | |
|---|---|---|---|---|---|---|---|---|---|---|---|
| | | | ABM | | | | Condition | | | | |
| | p-value | partial eta² | Negative Mean (SD) | Positive Mean (SD) | p-value | partial eta² | LL Mean (SD) | LH Mean (SD) | HL Mean (SD) | p-value | partial eta² |
| NUM (occurrences) | 0.207 | 0.06 | 3.6 (0.6) | 5.2 (0.5) | 0.053 | 0.14 | 5.1 (0.5) | 5.2 (0.5) | 2.9 (0.3) | <0.001* | 0.51 |
| P2P (deg/sec) | 0.573 | 0.02 | 0.57 (0.21) | 1.45 (0.16) | 0.003* | 0.30 | 1.09 (0.15) | 1.05 (0.13) | 0.90 (0.14) | 0.032* | 0.13 |
| AUC | 0.074 | 0.10 | 11.42 (6.07) | 37.11 (4.66) | 0.003* | 0.31 | 28.38 (4.88) | 26.41 (3.83) | 18.01 (3.30) | <0.001* | 0.31 |
| pAMP (microvolts) | 0.352 | 0.04 | 89.43 (13.13) | 91.77 (9.55) | 0.887 | 0.01 | 85.78 (9.77) | 91.10 (8.07) | 94.91 (8.75) | 0.352 | 0.04 |
| mFR (pps) | 0.795 | 0.01 | 5.10 (0.31) | 4.76 (0.22) | 0.374 | 0.03 | 4.64 (0.25) | 4.79 (0.18) | 5.35 (0.21) | <0.001* | 0.27 |

ABM = aberrant movement; NUM = number of zero-crossing; P2P = peak-to-peak of sudden deceleration and acceleration amplitude; AUC = area under sudden deceleration and acceleration curve; pAMP = peak motor unit amplitude; mFR = mean firing rate; LL = low speed low load; LH = low speed high load; HL = high speed low load; partial eta² = effect size partial eta-squared; SD = standard deviation

* = significant difference (p < 0.05)

Table 3 illustrates pairwise comparisons between groups and conditions for all kinematic and dEMG parameters. Post-hoc pairwise comparisons demonstrated significant between-group differences ($P < 0.05$) and exceeded the SEM in kinematic parameters, except for NUM, in LL and HL conditions ($P < 0.05$). However, the non-significant difference in NUM between groups in both conditions exceeded the SEM. Within-group comparisons did not show significant within-group changes ($P > 0.05$) between LL and LH in both groups for all kinematic parameters. We found significant changes ($P < 0.05$) comparing LL and HL conditions in both groups for NUM, but in IC group alone for P2P and AUC. In addition, changes in NUM and AUC exceeded the SEM, while changes in P2P did not exceed the SEM either group.

Although the results showed non-significant differences in pAMP, we wanted to further explore the motor unit behavior by performing comparisons using LSD (Table 3). No significant within-group changes ($P > 0.05$) were seen between LL and LH for pAMP and mFR in either group. When comparing LL and HL, the IC group demonstrated significant changes ($P < 0.05$) beyond the SEM in both pAMP and mFR. The group without IC findings for aberrant movement showed a significant change ($P < 0.05$) exceeding the SEM only in mFR. Table 4 presents a summary of individual profiles and potential strategies based on changes in motor unit behavior in response to each condition for each individual.

## 4. Discussion

Instability catch (IC) during active forward bending is an aberrant movement pattern in patients with low back pain. Increased load and speed may affect kinematics and motor unit behavior. This study sought to explore kinematic patterns across various loads and speeds while investigating motor unit behavior in individuals with and without IC.

Overall, kinematic results demonstrated individuals with IC had greater NUM, P2P and AUC than those without IC suggesting these parameters are representative of the presence of this aberrant movement pattern. Although the IC group did not show significantly greater NUM in the LL and HL conditions, this group still showed a trend toward greater NUM and the difference between groups exceeded the SEM. Our results were consistent with a previous study using kinematics that found a greater number of local minimum occurrences (comparable to NUM in our study) in individuals with aberrant movement [11]. Similar to the present findings, they found a mean difference between those with typical movement and IC was 2.4.

**Table 3. Mean and standard deviation for pairwise comparisons between negative and positive groups and among three conditions.**

| Parameter | SEM | ABM | Condition | | | LL vs LH | | | LL vs HL | | |
|---|---|---|---|---|---|---|---|---|---|---|---|
| | | | LL Mean (SD) | LH Mean (SD) | HL Mean (SD) | Diff | p-value | ES | Diff | p-value | ES |
| NUM (occurrences) | 0.7 | Negative | 4.4 (2.1) | 4.0 (1.3) | 2.4 (1.0) | 0.4 | 0.548 | 0.25 | 2.0[a] | 0.001* | 1.40 |
| | | Positive | 5.9 (2.9) | 6.3 (2.8) | 3.4 (1.8) | -0.4 | 0.301 | 0.20 | 2.5[a] | <0.001* | 1.36 |
| | | Diff | 1.5[a] | 2.3[a] | 1.0[a] | | | | | | |
| | | p-value | 0.172 | 0.024* | 0.105 | | | | | | |
| | | ES | 0.59 | 1.05 | 0.69 | | | | | | |
| P2P (deg/sec) | 0.35 | Negative | 0.66 (0.31) | 0.57 (0.24) | 0.49 (0.27) | 0.09 | 0.505 | 0.46 | 0.17 | 0.095 | 0.77 |
| | | Positive | 1.51 (0.94) | 1.54 (0.78) | 1.31 (0.86) | -0.03 | 0.804 | 0.06 | 0.20 | 0.014* | 0.57 |
| | | Diff | 0.85[a] | 0.97[a] | 0.82[a] | | | | | | |
| | | p-value | 0.011* | 0.001* | 0.007* | | | | | | |
| | | ES | 1.21 | 1.68 | 1.29 | | | | | | |
| AUC | 6.03 | Negative | 14.42 (8.53) | 11.55 (5.17) | 8.28 (4.87) | 2.87 | 0.511 | 0.48 | 6.14[a] | 0.111 | 0.97 |
| | | Positive | 42.34 (29.95) | 41.27 (23.72) | 27.73 (20.37) | 1.07 | 0.750 | 0.07 | 14.61[a] | <0.001* | 1.05 |
| | | Diff | 27.92[a] | 29.72[a] | 19.45[a] | | | | | | |
| | | p-value | 0.008* | 0.001* | 0.007* | | | | | | |
| | | ES | 1.27 | 1.79 | 1.31 | | | | | | |
| pAMP (microvolts) | 8.12 | Negative | 89.81 (71.37) | 88.21 (56.68) | 90.27 (53.73) | -1.6 | 0.897 | 0.03 | 0.46 | 0.962 | 0.04 |
| | | Positive | 81.74 (28.64) | 94.00 (26.37) | 99.55 (35.47) | 12.26[a] | 0.184 | 0.85 | 17.81[a] | 0.016* | 0.78 |
| | | Diff | -8.07 | 5.79 | 9.28[a] | | | | | | |
| | | p-value | 0.683 | 0.723 | 0.601 | | | | | | |
| | | ES | 0.15 | 0.13 | 0.20 | | | | | | |
| mFR (pps) | 0.37 | Negative | 4.84 (1.66) | 4.89 (0.98) | 5.57 (1.09) | 0.05 | 0.872 | 0.04 | 0.73[a] | 0.021* | 0.62 |
| | | Positive | 4.45 (0.93) | 4.69 (0.81) | 5.14 (0.96) | 0.24 | 0.323 | 0.33 | 0.69[a] | 0.011* | 0.98 |
| | | Diff | -0.39[a] | 0.20 | 0.43[a] | | | | | | |
| | | p-value | 0.443 | 0.573 | 0.310 | | | | | | |
| | | ES | 0.29 | 0.22 | 0.42 | | | | | | |

ABM = aberrant movement; NUM = number of zero-crossing; P2P = peak-to-peak of sudden deceleration and acceleration amplitude; AUC = area under sudden deceleration and acceleration curve; pAMP = peak motor unit amplitude; mFR = mean firing rate; SEM = Standard error of measurement; LL = low speed low load; LH = low speed high load; HL = high speed low load; SD = standard deviation; ES = effect size (Cohen's d).

* = significant difference (p < 0.05);

a = difference exceeds SEM

A potential limitation of measuring NUM alone is that it represents only the number of sudden decelerations and accelerations without consideration for the amplitude and timing of IC. These movement qualities can be quantified by P2P (taking amplitude into consideration) and AUC (taking amplitude and time into consideration). The same limitation measuring NUM may also be present with clinical observation of active forward bending in which small shaking (high NUM, but low amplitude) may not be detectable to the naked eye. Under the condition of increased load (LH), neither group showed significant changes in kinematic variables compared with the reference level. However, individuals with clinically observed IC had greater NUM, P2P, and AUC compared with those without IC. Additionally, effect sizes for group differences were amplified under the higher load condition. This may be the result of a greater challenge for the neuromuscular system [14]. Future investigations aiming to compare differences in kinematics in this population, might consider loaded conditions in order to elicit differences in task performance.

**Table 4. Summary of individual profiles and potential strategies.**

| Participant | Sex | HxLBP | Load | | | Speed | | |
|---|---|---|---|---|---|---|---|---|
| | | | Motor unit behavior | Kinematics | | Motor unit behavior | Kinematics | |
| | | | pAMP | mFR | AUC | pAMP | mFR | AUC |
| No Observed Instability Catch | | | | | | | | |
| dEMG01 | M | Y | ↓ | ↔ | ↔ | ↓ | ↑ | ↔ |
| dEMG04 | M | N | ↔ | ↔ | ↔ | ↔ | ↔ | ↔ |
| dEMG12 | M | N | ↑ | ↓ | ↔ | ↔ | ↔ | ↓ |
| dEMG16 | M | N | ↑ | ↔ | ↔ | ↑ | ↑ | ↔ |
| dEMG21 | M | Y | ↔ | ↔ | ↔ | ↔ | ↔ | ↔ |
| dEMG27 | M | N | ↔ | ↔ | ↔ | ↔ | ↔ | ↔ |
| dEMG07 | F | N | N/A | N/A | ↔ | N/A | N/A | ↔ |
| dEMG19 | F | Y | ↔ | ↑ | ↔ | ↔ | ↑ | ↔ |
| dEMG23 | F | N | ↔ | ↔ | ↔ | ↔ | ↑ | ↔ |
| dEMG26 | F | N | ↓ | ↑ | ↔ | ↔ | ↑ | ↔ |
| Sum | 4 | 3 | 2 | 2 | 0 | 1 | 5 | 1 |
| Percentage | 40.0%F | 30.0 | 22.2[a] | 22.2[a] | 0 | 11.1[a] | 55.6[a] | 14.3 |
| Observed Instability Catch | | | | | | | | |
| dEMG03 | M | N | ↑ | ↔ | ↑ | ↑ | ↔ | ↔ |
| dEMG08 | M | Y | ↔ | ↔ | ↔ | ↑ | ↔ | ↓ |
| dEMG09 | M | N | ↔ | ↔ | ↔ | ↔ | ↔ | ↓ |
| dEMG10 | M | N | ↔ | ↔ | ↔ | ↔ | ↔ | ↓ |
| dEMG11 | M | N | ↔ | ↑ | ↓ | ↔ | ↔ | ↓ |
| dEMG14 | M | Y | ↔ | ↔ | ↑ | ↑ | ↑ | ↔ |
| dEMG24 | M | Y | ↔ | ↑ | ↔ | ↔ | ↑ | ↔ |
| dEMG02 | F | Y | ↔ | ↔ | ↔ | ↔ | ↔ | ↔ |
| dEMG05 | F | N | ↑ | ↑ | ↔ | ↔ | ↔ | ↔ |
| dEMG06 | F | Y | ↔ | ↔ | ↔ | ↑ | ↔ | ↔ |
| dEMG13 | F | Y | ↔ | ↔ | ↓ | ↑ | ↑ | ↔ |
| dEMG15 | F | N | ↔ | ↔ | ↔ | ↔ | ↔ | ↓ |
| dEMG17 | F | Y | ↔ | ↑ | ↔ | ↔ | ↑ | ↓ |
| dEMG18 | F | Y | ↔ | ↔ | ↔ | ↑ | ↔ | ↔ |
| dEMG20 | F | Y | ↔ | ↔ | ↔ | ↔ | ↔ | ↔ |
| dEMG22 | F | Y | ↔ | ↔ | ↓ | ↔ | ↔ | ↓ |
| dEMG25 | F | Y | ↔ | ↔ | ↔ | ↔ | ↔ | ↔ |
| Sum | 10 | 11 | 2 | 4 | 3 | 6 | 4 | 7 |
| Percentage | 58.8%F | 64.7 | 11.8 | 23.5 | 17.6 | 35.3 | 23.5 | 41.2 |

M = male; F = female; HxLBP = history of low back pain; pAMP = peak motor unit amplitude; mFR = mean firing rate; AUC = area under sudden deceleration and acceleration curve;

[a] = based on 9 participants; Note: only increases were counted for dEMG parameters, and decreases were counted for kinematics parameter.

Within-group comparisons between LL and LH conditions indicate that increasing load did not impact movement patterns. Alternatively, significant decreases, larger than the SEM, were observed for NUM and AUC when speed was increased (HL); P2P also had a significant reduction in the IC group with increasing speed, but changes were within measurement error. This suggests changes in AUC could be due to changes in NUM, if P2P remained the same. Although the group without clinical signs of aberrant movement demonstrated a reduction trend, greater than the SEM, in AUC when increasing speed, these features might not be observed by clinicians because these values were less than baseline (LL condition).

Considering between group and between condition differences, it appears that AUC is the most sensitive kinematic measure.

Previous work by Orantes-Gonzalez et al. (2023) suggested that MU behavior responds differently to the conditions of speed and phase of the movement, with the concentric phase showing higher firing rates when compared to the eccentric and an increase in MU firing rates during the faster speed movements [15]. This information may allow an assessment of MU behavior and, depending on an individual's presentation, suggest whether faster or slower movements during different phases of movement within rehabilitation protocols are supported.

We did not find any changes in motor unit behavior when adding load. This is consistent with our findings from kinematics. However, changes were observed when increasing speed. These findings suggest that participants with and without IC used different strategies to perform the task where individuals without IC increased mFR, while individuals with IC increased pAMP and mFR. Theoretically, greater pAMP could be related to the recruitment of larger motor units [14,25]. Although these larger motor units can generate greater force, they are easily fatigable [25]. This could increase the risk of injury in situations that require repetitive and prolonged performance of a task. Motor unit behavior parameters are intercorrelated [14,17], therefore, we characterized individual responses to explore potential strategies for each group and investigated the effects of a history of LBP on these parameters (Table 3).

Differences in movement strategies between groups could be responsible for non-significant between-group differences when considering each motor unit behavior feature separately. This is evident only by exploring individual profiles. Individual subject profiles revealed that the IC group had a greater percentage of individuals with a history of LBP (64.7%) compared to those without IC (30%). Although they were asymptomatic at the time of testing, participants with a history of LBP continued to present with aberrant movement. This finding is consistent with previous reports where aberrant movements were observed after the resolution of pain or following an episode of LBP [1,6]. Aberrant movements can cause alterations in normal shear forces and suboptimal tissue loading thereby increasing the risk of re-injury in those with a history of LBP [4,5]. According to our motor unit behavior findings, it seems that these aberrant movements could be caused by altered motor unit behavior. Approximately twenty-two percent of individuals without IC increased pAMP and mFR as a strategy to perform forward bend with higher load, while 23.5% of the IC group increased mFR during this condition. When increasing speed, a majority (55.6%) of individuals without IC increased mFR as a strategy, which is consistent with the results presented in Table 3. In contrast, a greater percentage (35.3%) of individuals with IC relied on increased pAMP to accommodate faster forward bending. Motor unit firing rate strategy in individuals without IC were consistent with strategies commonly reported in dEMG studies [15,26]. This finding suggests that when individuals are required to perform faster movements, the neuromuscular control system increases motor unit firing rate to match the demand being placed on the neuromuscular control system [15].

The IC group demonstrated a smaller percentage of individuals who increased mFR in response to the condition with increased speed. Some individuals with IC (dEMG08 and 17) utilized motor unit behavior strategies that were sufficient to improve movement control as evidenced by decreased AUC during this condition, while other participants in the IC group (dEMG 03, 14, 24, 06, 13, 18) demonstrated changes in motor unit behavior, but these strategies were insufficient to impact AUC. Additionally, participants 9, 10, 11, 15, 22 did not demonstrate any changes in motor unit behavior variables, while decreasing AUC during this condition. It is worth noting that four of these participants had no history of LBP, which may have been sufficient to improve movement control. It is also important to consider that active forward bending requires significant contribution from muscle groups such as the erector

spinae and gluteal muscles, in addition to the LM. Evidence supports altered muscle activation patterns in these muscle groups in patients with LBP [3,25,27]. Future studies should include a wider array of trunk and hip muscles to better understanding motor unit behaviors in this population.

Although statistical analysis found a non-significant between-group difference in mean velocity across conditions, individuals with IC group moved slightly faster than those without IC. This might suggest the inability to control the movement during forward bend. To the best of our knowledge, this is the first study to investigate motor unit behavior response to conditions of increased load and speed. Therefore, we are unable to compare our results with other studies. However, one study reported on the motor unit behavior of females analyzing dominant and non-dominant back muscles [22]. They reported greater mFR than our findings. The different values could reflect differences in protocols. They collected data from the lumbar erector spinae using a 40% bodyweight load which resulted in a greater neuromuscular requirement. Additionally, the lumbar erector spinae primarily function as a force generator for trunk movement, which should be composed of larger motor units and the ability to generate force at higher mFR [28].

Our study has some limitations. First, clinical observation and kinematics were collected during the eccentric phase of movement, while dEMG considered both eccentric and concentric contractions. Our study investigated only the LM muscles; therefore, the motor unit behavior changes in our study are not generalizable to other muscles. Future studies should include other muscles involved in this forward bend task. At this time, the kinematic metrics identified are associated with clinical observations, and further validation with different groups with and without instability catch is required. In addition, we aimed to explore motor unit behavior in response to increased load and speed. Our primary statistical analysis did not show significant findings; however, our detailed exploration of group and individual movement profiles should inform the design of future studies. Lastly, altered motor unit behavior could result from mechanical joint dysfunction, such as facet joint hypomobility, and this faulty joint movement may be related to an aberrant movement pattern. Therefore, future studies should include a comprehensive examination by experienced clinicians to identify other possible underlying causes of IC (e.g., facet joint dysfunction, intervertebral joint impairments, etc.).

## 5. Conclusion

Kinematic results demonstrated individuals with IC had greater NUM, P2P and AUC than those without IC suggesting these parameters can represent IC. Changes in motor unit behavior were found when increasing speed and these changes were different between individuals with and without IC suggesting these groups used a different neuromuscular strategy to perform the task.

## Supporting information

**S1 File. Minimal anonymized data set.**
(XLSX)

## Acknowledgement

We would like to thank Mahidol University for providing travel grant (Academic and staff mobility program). We also would like to thank the Spine Biomechanics Laboratory, Mahidol University, for providing data collection space and equipment. We also would like to express our appreciation to all participants who joined in this study.

## Author contributions

**Conceptualization:** Peemongkon Wattananon.

**Data curation:** Peemongkon Wattananon, Sasithorn Kongoun, Katayan Klahan.

**Formal analysis:** Peemongkon Wattananon, Sheri P. Silfies, John R. Gilliam.

**Funding acquisition:** Peemongkon Wattananon.

**Methodology:** Peemongkon Wattananon, Sasithorn Kongoun, Katayan Klahan.

**Writing – original draft:** Peemongkon Wattananon, Sheri P. Silfies, John R. Gilliam, Jim Richards.

**Writing – review & editing:** Peemongkon Wattananon, Sheri P. Silfies, John R. Gilliam, Jim Richards.

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
