## [Decision Letter · Decision Letter 0]

13 Jan 2025

PONE-D-24-57098Trunk kinematics and motor unit behavior during different loads and speeds in individuals with and without aberrant movement patterns during active forward bending: A cross-sectional studyPLOS ONE

Dear Dr. Wattananon,

Thank you for submitting your manuscript to PLOS ONE. After careful consideration, we feel that it has merit but does not fully meet PLOS ONE’s publication criteria as it currently stands. Therefore, we invite you to submit a revised version of the manuscript that addresses the points raised during the review process.

We look forward to receiving your revised manuscript.

Kind regards,

Seyed Hamed Mousavi

Academic Editor

PLOS ONE

Journal Requirements:

Reviewers' comments:

Reviewer's Responses to Questions

**Comments to the Author**

1. Is the manuscript technically sound, and do the data support the conclusions?

Reviewer #1: Yes

Reviewer #2: Yes

2. Has the statistical analysis been performed appropriately and rigorously? 

Reviewer #1: Yes

Reviewer #2: I Don't Know

3. Have the authors made all data underlying the findings in their manuscript fully available?

Reviewer #1: Yes

Reviewer #2: Yes

4. Is the manuscript presented in an intelligible fashion and written in standard English?

Reviewer #1: Yes

Reviewer #2: Yes

5. Review Comments to the Author

Reviewer #1: This quasi-experimental study appears to be needed, which was provided by your introduction; clear purpose and hypothesis statements. The methods requires some additional information. The results are clear and concisely presented. The discussion and conclusion related to the problem identified, most important results, and primary purpose. See pdf for specific comments for methods, results, and the discussion. Some word tense, etc. errors were highlighted.

Reviewer #2: The paper investigates trunk kinematics and motor unit behavior during forward bending tasks performed at varying speeds and loads in individuals with and without aberrant movement patterns, known as "instability catch" (IC). Using inertial measurement units (IMUs) and decomposition electromyography (dEMG), the study identifies differences in movement strategies between the groups. Key findings highlight that individuals with IC exhibit greater amplitude and area of trunk deceleration/acceleration, while neuromuscular control strategies differ, as those with IC rely more on peak motor unit amplitudes (pAMP) and less on mean firing rates (mFR) under increased speed. These findings suggest the potential of kinematic and motor unit behavior metrics for characterizing IC and inform clinical strategies for improving trunk control.

The authors are commended for their rigorous methodology, use of innovative dEMG technology, and exploration of clinically relevant movement patterns. To further strengthen the paper, the following revisions are suggested:

Expand on how kinematic metrics such as NUM, P2P, and AUC could be validated as objective clinical markers for IC detection, potentially aiding in the standardization of IC assessment across broader clinical populations.

Discuss in greater depth how increased speed influences motor unit recruitment and firing rate, particularly in individuals with IC, and how this could inform the development of speed-based neuromuscular training protocols.

Explore the feasibility of using IMU and dEMG systems for real-time monitoring and feedback in clinical or rehabilitation settings, emphasizing their potential to improve movement control and reduce aberrant patterns dynamically.

6. PLOS authors have the option to publish the peer review history of their article (what does this mean? ). If published, this will include your full peer review and any attached files.

**Do you want your identity to be public for this peer review?** For information about this choice, including consent withdrawal, please see our Privacy Policy .

Reviewer #1: No

Reviewer #2: No

---

## [Author Response · Author response to Decision Letter 0]

30 Jan 2025

Reviewer #1:

1. This quasi-experimental study appears to be needed, which was provided by your introduction; clear purpose and hypothesis statements. The methods requires some additional information. The results are clear and concisely presented. The discussion and conclusion related to the problem identified, most important results, and primary purpose. See pdf for specific comments for methods, results, and the discussion. Some word tense, etc. errors were highlighted.

Response: Thank you for your valuable feedback, which has helped us strengthen our manuscript. We also word tense errors highlighted throughout the manuscript.

2. “Two inertial measurement unit (IMU) sensors” Please describe pre-testing calibration of the IMUs, including establishment of laboratory coordinate system, etc; with references

Response: Thank you for your comment. We added and revised this part as following.

“Two Inertial Measurement Unit (IMU) sensors (Trigno Avanti, Delsys Inc., MA, USA) were attached to the lumbar (L1) and sacral (S2) spinous processes. Angular velocity data were recorded directly from the gyroscopes of the IMUs during an active forward bend movement at 370 Hz. As these were carefully placed on anatomical segments and no further parameters were calculated, these required no additional calibration process [19, 20]. The sensor axes were used to define the planes of movement in rotation about X (flexion/extension), Y (rotation), and Z (lateral bending) of the lumbar and sacral segments in which flexion, rotation to the right, and lateral bend to the left were considered positive directions (Figure 1A). This method has been previously used to explore lumbopelvic movements [19, 20], and angular velocity measures have shown excellent test-retest reliability to assess movement pattern consistency (coefficient of multiple determination = 0.85) [20] and sensitivity changes between conditions [21].”

References

- Budini K, Richards J, Cole T, et al. An exploration of the use of Inertial Measurement Units in the assessment of dynamic postural control of the knee and the effect of bracing and taping. Physiother Pract Res. 2018;39:91-98.

- Wattananon P, Kongoun S, Chohan A, Richards J. The use of statistical parametric mapping to determine altered movement patterns in people with chronic low back pain. J Biomech. 2023;153:111601.

- Brabants A, Richards J, Deschamps K, Janssen J, Chohan A, Connell L. An exploration of segment acceleration and angular velocity during different balance conditions measures in the assessment of stability. PRM+ J Quant Res Rehabil Med. 2018;1(2):30-6.

3. “Two dEMG sensors (Trigno Galileo, Delsys Inc., MA, USA) were attached bilaterally to the lumbar multifidus (2 cm lateral to L4 spinous process) with the reference attached over the iliac crests. These each collected four channels of EMG data from 4 protruding blunted pins with 5-mm inter-pin space.” More information is needed when reporting research using EMG. See: Merletti, R. Standards for reporting EMG data. J Electromyogr Kinesiol. 1999, 9(1), 3-4.

Response: We added information as follows.

Instruments and measures

“Each sensor collected four channels of differential EMG data at 2222 Hz from 4 protruding blunted pins (0.5 mm in diameter) with 5-mm inter-pin space. EMG data were amplified with a gain of 1000 and filtered at a band-pass of 20-450 Hz [15, 18, 22].”

Procedure

“Skin preparation was performed using a 70% alcohol swab prior to IMU and dEMG sensor attachment (Figure 1A). Participants were asked to relax in the prone position and baseline noise was assessed to ensure the value was less than ±10 microvolts for the dEMG data. If the baseline noise was greater than ±10 microvolts this process was repeated until this baseline noise level was achieved.”

References

- Orantes-Gonzalez E, Heredia-Jimenez J, Lindley SB, Richards JD, Chapman GJ. An exploration of the motor unit behaviour during the concentric and eccentric phases of a squat task performed at different speeds. Sports Biomech. 2023:1-12.

- Nawab SH, Chang SS, De Luca CJ. High-yield decomposition of surface EMG signals. Clin Neurophysiol. 2010;121(10):1602-15.

- Silva MF, Dias JM, Pereira LM, et al. Determination of the motor unit behavior of lumbar erector spinae muscles through surface EMG decomposition technology in healthy female subjects. Muscle Nerve. 2017;55(1):28-34.

- Merletti, R. Standards for reporting EMG data. J Electromyogr Kinesiol. 1999, 9(1), 3-4.

4. “dEMG data processing was performed using the Neuromap software” Please describe any filtering that was done; provide references

Response: For dEMG data processing, no filtering was required. The program uses raw dEMG data and decomposes into individual motor unit action potential trains using an artificial intelligence algorithm. We revised this part as follows.

“For dEMG data processing, no filtering was required. NeuroMap software version XXXX (Delsys, Inc., Boston, USA) was used to decompose the EMG signals into individual motor units using an artificial intelligence algorithm [18]. Neuromap Explorer (Delsys, Inc., Boston, USA) was then used to extract the peak motor unit amplitude (pAMP) and mean firing rates (mFR) that had an accuracy of 80% or greater, which is supported by De Luca et al. [16], who demonstrated that 80% is appropriate for identifying a comparable number of motor unit action potential trains during dynamic, cyclic tasks.”

References

- De Luca CJ, Chang SS, Roy SH, Kline JC, Nawab SH. Decomposition of surface EMG signals from cyclic dynamic contractions. J Neurophysiol. 2015;113(6):1941-51.

- De Luca CJ, Contessa P. Biomechanical benefits of the Onion-Skin motor unit control scheme. J Biomech. 2015;48(2):195-203.

- Nawab SH, Chang SS, De Luca CJ. High-yield decomposition of surface EMG signals. Clin Neurophysiol. 2010;121(10):1602-15.

5. “kinematic and dEMG” please provide a list of all dependent variables; both full spelling along with acronyms; this will make it easier for the reader when they look at results and discussion. For example, when I see NUM I have no idea what this is.

Response: We added a list of dependent variables as follows.

“Kinematic parameters include mean angular velocity (MV), peak angular velocity (PV), number of acceleration zero-crossings (NUM), peak-to-peak acceleration amplitude (P2P), and total area under sudden deceleration and acceleration curves (AUC). dEMG parameters include peak motor unit amplitude (pAMP) and mean firing rates (mFR).”

6. “We used a two-factor mixed model ANOVA with post-hoc least significant difference (LSD) and SEM to determine interaction and main effects of group (presence and absence of IC) and condition (LL, LH and HL).” Please also describe the determination of effect sizes, the scale used to rate the magnitude of effect sizes; please reference

Response: We added the details as follows.

“Effect sizes of two-factor mixed model ANOVA were reported using partial eta squared (partial 2) and interpreted as small (0.01), medium (0.06), and large (0.14), while effect sizes of pairwise comparison were reported using Cohen’s d and interpreted as small (0.3), medium (0.5), and large (0.8) [24].”

Reference

- Cohen J. Statistical Power Analysis for the Behavioral Sciences. 2nd ed: Routledge; 1988.

7. Discussion section: Before you summarize results, please: 1) briefly restate the problem identified and rationale/need for this study and 2) the primary purpose of the study.

Response: We added details as the reviewer suggested as follows.

“Instability catch (IC) during active forward bending is an aberrant movement pattern in patients with low back pain. Increased load and speed may affect kinematics and motor unit behavior. This study sought to explore kinematic patterns across various loads and speeds while investigating motor unit behavior in individuals with and without IC.”

8. “According to our motor unit behavior findings, it seems that these aberrant movement could be caused by altered motor unit behavior.” motor unit behavior is often a response to a mechanical fault and not the primary fault; for example, for those of us who treat persons with back pain we know that abberent movement patterns are related to abnormal spinal unit mechanics related to facet joint and intervertebral joint impairments; manipulative therapy restores normal kinematics and secondarily restores normal motor function; however, these faulty joint kinematics can return with subsequent minor traumas, e.g., sometimes just activities of daily living, resulting also then in a return of altered motor unit behavior. Thus, there is a complex interplay between spine joint function and muscle function that needs to be considered in your discussion. You might explore the literature on what I have summarized.

Response: We agree with the reviewer in that changes in motor unit behavior could be a response to mechanical joint dysfunction. Hypomobility of the intervertebral joint can cause aberrant movement. We added this information as one of our limitations.

“Lastly, altered motor unit behavior could result from mechanical joint dysfunction, such as facet joint hypomobility, and this faulty joint movement may be related to an aberrant movement pattern.”

9. “Future studies should include a wider array of trunk and hip muscles to better understanding motor unit behaviors in this population.” future studies need to include a thorough examination for impaired spinal unit movements of participants by a skilled manual physical therapist.

Response: We added the following sentence.

“Therefore, future studies should include a comprehensive examination by experienced clinicians to identify other possible underlying causes of IC (e.g., facet joint dysfunction, intervertebral joint impairments, etc.).”

Reviewer #2:

The paper investigates trunk kinematics and motor unit behavior during forward bending tasks performed at varying speeds and loads in individuals with and without aberrant movement patterns, known as "instability catch" (IC). Using inertial measurement units (IMUs) and decomposition electromyography (dEMG), the study identifies differences in movement strategies between the groups. Key findings highlight that individuals with IC exhibit greater amplitude and area of trunk deceleration/acceleration, while neuromuscular control strategies differ, as those with IC rely more on peak motor unit amplitudes (pAMP) and less on mean firing rates (mFR) under increased speed. These findings suggest the potential of kinematic and motor unit behavior metrics for characterizing IC and inform clinical strategies for improving trunk control.

The authors are commended for their rigorous methodology, use of innovative dEMG technology, and exploration of clinically relevant movement patterns. To further strengthen the paper, the following revisions are suggested:

1. Expand on how kinematic metrics such as NUM, P2P, and AUC could be validated as objective clinical markers for IC detection, potentially aiding in the standardization of IC assessment across broader clinical populations.

Response: Thank you for your valuable suggestions. This is an important point to tell the readers. We have acknowledged that further validation is required in the limitations section.

“At this time, the kinematic metrics identified are associated with clinical observations, and further validation with different groups with and without instability catch is required.”

2. Discuss in greater depth how increased speed influences motor unit recruitment and firing rate, particularly in individuals with IC, and how this could inform the development of speed-based neuromuscular training protocols.

Response: Thank you for your suggestion. We added details in the discussion section as follows.

“Previous work by Orantes-Gonzalez et al. (2023) suggested that MU behavior responds differently to the conditions of speed and phase of the movement, with the concentric phase showing higher firing rates when compared to the eccentric and an increase in MU firing rates during the faster speed movements [15]. This information may allow an assessment of MU behavior and, depending on an individual’s presentation, suggest whether faster or slower movements during different phases of movement within rehabilitation protocols are supported.”

3. Explore the feasibility of using IMU and dEMG systems for real-time monitoring and feedback in clinical or rehabilitation settings, emphasizing their potential to improve movement control and reduce aberrant patterns dynamically.

Response: At this time real-time dEMG is not possible and requires time consuming analysis, however further developments may allow this in the future. Assessment of quality of movement however can be assessed quickly using IMUs and the methods described in this paper. These do not require any additional calibration process or additional signal processing, which was part of the rationale of the authors to utilize this approach. Such uses of IMUs show great promise to offer fast and simple measures which could be implemented during existing movements used within clinical assessments.

---

## [Decision Letter · Decision Letter 1]

23 Feb 2025

PONE-D-24-57098R1Trunk kinematics and motor unit behavior during different loads and speeds in individuals with and without aberrant movement patterns during active forward bending: A cross-sectional studyPLOS ONE

Dear Dr. Wattananon,

Thank you for submitting your manuscript to PLOS ONE. After careful consideration, we feel that it has merit but does not fully meet PLOS ONE’s publication criteria as it currently stands. Therefore, we invite you to submit a revised version of the manuscript that addresses the points raised during the review process.

We look forward to receiving your revised manuscript.

Kind regards,

Seyed Hamed Mousavi

Academic Editor

PLOS ONE

Journal Requirements:

Reviewers' comments:

Reviewer's Responses to Questions

**Comments to the Author**

1. If the authors have adequately addressed your comments raised in a previous round of review and you feel that this manuscript is now acceptable for publication, you may indicate that here to bypass the “Comments to the Author” section, enter your conflict of interest statement in the “Confidential to Editor” section, and submit your "Accept" recommendation.

Reviewer #1: All comments have been addressed

2. Is the manuscript technically sound, and do the data support the conclusions?

Reviewer #1: Yes

3. Has the statistical analysis been performed appropriately and rigorously? 

Reviewer #1: Yes

4. Have the authors made all data underlying the findings in their manuscript fully available?

Reviewer #1: Yes

5. Is the manuscript presented in an intelligible fashion and written in standard English?

Reviewer #1: Yes

6. Review Comments to the Author

Reviewer #1: With the revised content, the authors made a good paper better. I made some additional comments on minor word tense and word choice errors, which can be found in the revised pdf.

7. PLOS authors have the option to publish the peer review history of their article (what does this mean? ). If published, this will include your full peer review and any attached files.

**Do you want your identity to be public for this peer review?** For information about this choice, including consent withdrawal, please see our Privacy Policy .

Reviewer #1: No

---

## [Author Response · Author response to Decision Letter 1]

24 Feb 2025

Reviewers' comments:

Reviewer's Responses to Questions

Comments to the Author

1. If the authors have adequately addressed your comments raised in a previous round of review and you feel that this manuscript is now acceptable for publication, you may indicate that here to bypass the “Comments to the Author” section, enter your conflict of interest statement in the “Confidential to Editor” section, and submit your "Accept" recommendation.

Reviewer #1: All comments have been addressed

2. Is the manuscript technically sound, and do the data support the conclusions?

Reviewer #1: Yes

3. Has the statistical analysis been performed appropriately and rigorously?

Reviewer #1: Yes

4. Have the authors made all data underlying the findings in their manuscript fully available?

Reviewer #1: Yes

5. Is the manuscript presented in an intelligible fashion and written in standard English?

Reviewer #1: Yes

6. Review Comments to the Author

Reviewer #1: With the revised content, the authors made a good paper better. I made some additional comments on minor word tense and word choice errors, which can be found in the revised pdf.

Response: Thank you for your valuable feedback. We corrected the word tense and word choice errors highlighted throughout the manuscript.

7. PLOS authors have the option to publish the peer review history of their article (what does this mean?). If published, this will include your full peer review and any attached files.

Do you want your identity to be public for this peer review? For information about this choice, including consent withdrawal, please see our Privacy Policy.

Reviewer #1: No

---

## [Editor Report · Decision Letter 2]

2 Mar 2025

Trunk kinematics and motor unit behavior during different loads and speeds in individuals with and without aberrant movement patterns during active forward bending: A cross-sectional study

PONE-D-24-57098R2

Dear Dr. Wattananon,

We’re pleased to inform you that your manuscript has been judged scientifically suitable for publication and will be formally accepted for publication once it meets all outstanding technical requirements.

Kind regards,

Seyed Hamed Mousavi

Academic Editor

PLOS ONE
---

## [Editor Report · Acceptance letter]

PONE-D-24-57098R2

PLOS ONE

Dear Dr. Wattananon,

I'm pleased to inform you that your manuscript has been deemed suitable for publication in PLOS ONE. Congratulations! Your manuscript is now being handed over to our production team.

Kind regards,

on behalf of

Dr. Seyed Hamed Mousavi

Academic Editor

PLOS ONE